# Calpain-1 and Calpain-2 in the Brain: New Evidence for a Critical Role of Calpain-2 in Neuronal Death

**DOI:** 10.3390/cells9122698

**Published:** 2020-12-16

**Authors:** Yubin Wang, Yan Liu, Xiaoning Bi, Michel Baudry

**Affiliations:** 1Graduate College of Biomedical Sciences, Western University of Health Sciences, Pomona, CA 91766, USA; wangy1@westernu.edu (Y.W.); yliu@westernu.edu (Y.L.); 2College of Osteopathic Medicine of the Pacific, Western University of Health Sciences, Pomona, CA 91766, USA; xbi@westernu.edu

**Keywords:** learning, neurodegeneration, hippocampus, signaling pathways

## Abstract

Calpains are a family of soluble calcium-dependent proteases that are involved in multiple regulatory pathways. Our laboratory has focused on the understanding of the functions of two ubiquitous calpain isoforms, calpain-1 and calpain-2, in the brain. Results obtained over the last 30 years led to the remarkable conclusion that these two calpain isoforms exhibit opposite functions in the brain. Calpain-1 activation is required for certain forms of synaptic plasticity and corresponding types of learning and memory, while calpain-2 activation limits the extent of plasticity and learning. Calpain-1 is neuroprotective both during postnatal development and in adulthood, while calpain-2 is neurodegenerative. Several key protein targets participating in these opposite functions have been identified and linked to known pathways involved in synaptic plasticity and neuroprotection/neurodegeneration. We have proposed the hypothesis that the existence of different PDZ (PSD-95, DLG and ZO-1) binding domains in the C-terminal of calpain-1 and calpain-2 is responsible for their association with different signaling pathways and thereby their different functions. Results with calpain-2 knock-out mice or with mice treated with a selective calpain-2 inhibitor indicate that calpain-2 is a potential therapeutic target in various forms of neurodegeneration, including traumatic brain injury and repeated concussions.

## 1. Introduction

Calcium-activated neutral proteases (CANPs) were discovered in 1964 by Guroff [1], but in 1980 Murachi changed their name to calpain, as a contraction of calpain and papain, and also coined the name calpastatin for their endogenous inhibitor [2]. Since then, many studies have been directed at understanding the physiological as well as the pathological function(s) of this family of proteases in the brain and other organs. We initially proposed in 1984 that calpain played a critical role in long-term potentiation (LTP) and learning and memory [3]. This hypothesis was recently validated by studies performed first in hippocampal slices from calpain-4 knock-out (KO) mice [4] and later in slices from calpain-1 KO mice [5,6]. Following our initial studies on the potential role of calpain in LTP and learning and memory, a large number of studies focused on the potential role of calpain in neuronal death and neurodegeneration [7,8,9,10,11,12,13,14]. During the same period, a plethora of calpain isoforms were identified and we now know that calpains constitute a family of enzymes with at least 15 members [15,16], with calpain-1 (aka μ-calpain) and calpain-2 (aka m-calpain) being the most ubiquitous isoforms in all tissues and organs, including the brain. While there is strong evidence that calpain plays a role in neurodegeneration, there are only a handful of studies addressing the question of which calpain isoform(s) is (are) involved and of which are the signaling pathways leading to neurodegeneration. Genetic studies have provided some information regarding the potential contributions of various calpain isoforms in human diseases [17]. In particular, defects in the gene encoding the muscle-specific calpain-3 lead to a particular type of dystrophy, limb-girdle muscular dystrophy 2A (LGMD-2A) [18,19]. There is also good evidence for a link between calpain-10 and diabetes mellitus, based on genetic studies [20]. More recently, calpain-14 has been linked to eosinophilic esophagitis, due to its abundance in the upper gastro-intestinal tract [21]. Mutations in calpain-5 have recently been associated with autoimmune uveitis and photoreceptor degeneration [22]. Very recent studies have also linked mutations in calpain-15 with various types of developmental eye disorders in humans [23]. Our laboratory has focused mostly on the study of the roles of calpain-1 (aka, µ-calpain) and calpain-2 (aka m-calpain) in the brain [14]. Calpain-1 and calpain-2 exhibit a high degree (>70%) of homology. Early in vitro biochemical studies suggested the major difference consisted in their calcium requirement for activation, with calpain-1 requiring micromolar calcium concentrations and calpain-2 millimolar calcium concentrations [2]. This presented a significant challenge to study the potential role of calpain-2 in the brain, as such a high calcium requirement for calpain-2 activation made it unlikely that cytoplasmic calpain-2 could be activated under physiological and most pathological conditions. We were thus left with the paradox of explaining how the same enzyme, calpain-1, could be involved in both synaptic plasticity and neurodegeneration. This review will summarize our work over the last 20–25 years directed at resolving this paradox. What we found turned out to be quite remarkable: in short, our studies revealed that calpain-1 and calpain-2 play opposite functions in the brain, with calpain-1 activation being required for triggering certain forms of synaptic plasticity and thereby in various forms of learning and memory; in addition, calpain-1 is neuroprotective both during postnatal development and in adulthood. In contrast, calpain-2 activation limits the magnitude of synaptic plasticity and the extent of learning and is neurodegenerative [14]. These different functions of calpain-1 and calpain-2 and the signaling pathways they regulate to perform these functions will be discussed in greater details in this review.

## 2. Calpain-1 Role in Synaptic Plasticity

As mentioned above, the hypothesis that calpain plays a significant role in LTP and learning and memory was proposed in 1984 [3]. Over the following 10 years, several findings indicated that calpain-1 and calpain-2, by cleaving several key proteins, participated in the regulation of dendritic structure and local protein synthesis. The first hint that calpain was linked to the regulation of local protein translation was provided by a report that calpain cleaved dicer and released dicer and eIF2c from postsynaptic densities, thereby facilitating the processing of miRNA and regulating protein translation [24]. It was later reported that calpain could cleave and inactivate the suprachiasmatic nucleus circadian oscillatory protein (SCOP, also known as PH domain and leucine-rich repeat protein phosphatase 1 (PHLPP1β)) [25], a negative regulator of the extracellular signal-regulated kinase (ERK), a kinase with numerous links to LTP [26], thus linking calpain activation to activation of the ERK pathway. Calpain was also shown to cleave β-catenin, generating an active fragment, which regulates gene transcription, thereby providing a mechanism by which NMDA (N-Methyl-D-Aspartate) receptor stimulation, which has been repeatedly linked to calpain activation [27,28], could modify gene expression [29]. We also found that several scaffolding proteins were calpain substrates, including PSD95 [30], GRIP [31], SAP97 [32], and more recently ankyrin repeat-rich membrane spanning protein (ARMS) or kinase D-interacting substrate of 220 kDa (Kidins220) was shown to be regulated by calpain-mediated truncation [33]. By degrading the translational repressor poly(A)-binding protein (PABP)-interacting protein 2A (PAIP2A), an inhibitor of PABP, calpain could also relieve translational inhibition of proteins, a mechanism that has been involved in synaptic plasticity and learning and memory [34]. 

More recently, the role of calpain in long-term potentiation (LTP) was supported by the study using hippocampal slices from mice with a conditional downregulation of calpain-4, the small subunit required by both calpain-1 and calpain-2 for functional activity. Theta burst stimulation (TBS)-induced LTP induction was impaired in hippocampal slices prepared from these mice [4]. 

Interestingly, other targets of calpain also link calpain activation to some of the molecular/cellular mechanisms known to participate in LTP; in particular, we identified a pathway linking actin polymerization in dendritic spines to the synaptic structural modifications associated with LTP [35]. Three actin-signaling pathways involving the Rho family of small GTPases, RhoA, Rac, and Cdc42, are ubiquitously implicated in the mechanisms of actin filament assembly, disassembly, or stabilization in most cells [36], and have been shown to be critically involved in LTP consolidation [37,38]. We discovered that, like PHLPP1β, RhoA is rapidly degraded and then resynthesized after LTP through the sequential activation of calpain-1 and calpain-2 [39].

The results obtained with calpain inhibitors or the calpain-4 KO mice were further confirmed by experiments showing that calpain-1 KO mice were impaired in TBS-LTP and in hippocampus-dependent learning [6], clearly indicating that calpain-1 activation following synaptic NMDA receptor stimulation is essential for theta burst stimulation-induced LTP in field CA1 of the hippocampus. More recently, we also found that calpain-1 activation was required for mGluR-dependent long-term depression (LTD) in hippocampus [40]. The molecular pathway involved in this case is the inactivation of PP2A due to the cleavage of B56α, a regulatory subunit of PP2A. This leads to mTOR stimulation and a local increase in the synthesis of activity-regulated cytoskeleton-associated protein (Arc) and reduced levels of GluA1-containing a-amino-3-hydroxy-5-methyl-4-isoxazolepropionic acid (AMPA) receptors. In support of this finding, LTD impairment in calpain-1 KO mice was rescued by PP2A inhibitors [40]. We also found that calpain-1 was involved in various forms of LTD in the cerebellum, and the molecular pathways involved were quite similar to those we found in the hippocampus. Thus, dihydroxyphenylglycine (DHPG)-LTD was enhanced by the application of okadaic acid, a PP2A inhibitor, in cerebellar slices from both WT and calpain-1 KO mice. These results indicated that the same signaling pathway, calpain-1-→PP2A-→Arc-→AMPA receptor internalization, is involved in both hippocampal and cerebellar LTD [41]. Previous studies had reported that cerebellar LTD required the dephosphorylation of transmembrane AMPAR regulatory protein γ-2 (TARP γ-2), a process known to result in AMPA receptor internalization [42]. As we also found that TARP γ-2 is a calpain substrate [43], we proposed that calpain-mediated truncation of TARP γ-2 contributes to cerebellar LTD. 

Over the last 20 years, LTP and LTD at the parallel fiber to Purkinje cell synapses have been proposed to participate in the learning of various cerebellum-dependent tasks, including classical conditioning of motor responses [44,45,46], although some evidence suggests that LTD at the parallel fiber to Purkinje cell synapses is not required for cerebellar motor learning [47]. In agreement with the role of calpain-1 in LTD in the cerebellum, we found that calpain-1 KO mice were impaired in the initial acquisition of classically conditioned eye-blink responses, although they were able to reach the same level of performance as the WT mice with continued training [41]. In addition, calpain-1 KO mice did not show any impairment during extinction of classical conditioning eye-blink responses.

Overall, all the studies reported above clearly indicate that calpain-1 plays an important role in various forms of synaptic plasticity and related types of learning and memory. Some of the molecular pathways regulated by calpain-1 activation in these mechanisms have been identified and these findings might lead to future potential treatments for diseases associated with learning and memory impairment.

## 3. Calpain-1 Role in Neuroprotection

The first hint that calpain-1 activation was neuroprotective came from studies showing that calpain-1 could cleave PHLPP1, resulting in Akt activation, a known pathway leading to neuronal survival [48]. This idea was further confirmed by demonstrating that calpain-1-mediated truncation of PHLPP1 and Akt activation was involved in NMDAR-dependent survival of cerebellar granule cells (CGCs) [49]. Later studies indicated that calpain-1 KO mice exhibited abnormal cerebellar development, including enhanced apoptosis of CGCs during the early postnatal period, reduced granule cell density and impaired synaptic transmission from the parallel fibers to Purkinje cells, resulting in cerebellar ataxia [49]. All these defects were due to deficits in the calpain-1/PHLPP1/Akt pro-survival pathway in developing granule cells, since treatment with an Akt activator during the postnatal period or crossing calpain-1 KO mice with PHLPP1 KO mice restored most of the observed alterations in cerebellar structure and function in calpain-1 KO mice [49]. Interestingly, several human families carrying CAPN1 mutations were found to exhibit cerebellar ataxia [49,50] and Russell terrier dogs carrying a missense mutation in calpain-1 also exhibit spinocerebellar ataxia [51]. 

To further demonstrate the role of calpain-1 in neuroprotection, we determined the extent of neuronal damage elicited by acute insults in wild-type mice and calpain-1 KO mice. We postulated that the extent of damage would be greater in the KO mice than in the WT littermates. Thus, we found that, following traumatic brain injury (TBI), cell death and lesion volume were larger in calpain-1 KO mice than in WT mice [52]. In addition, both retinal ganglion cell death following acute increased intraocular pressure and hippocampal neuronal death following kainate-induced seizure were enhanced in calpain-1 KO mice, as compared to WT mice [53,54]. Calpain deletion also results in neuronal degeneration across different species including human, dog, mice, zebrafish, fruit fly, and Caenorhabditis elegans [49,51,55]. All these results suggest a neuroprotective role for calpain-1 both during the postnatal period and in the adult.

## 4. Calpain-2 Role in Synaptic Plasticity

While the studies discussed above clearly supported a necessary role for calpain-1 activation in the induction phase of LTP, very little information was available concerning the potential role of calpain-2 activation in synaptic plasticity and in LTP. As mentioned, a key issue was the high calcium concentration required for calpain-2 activation, and there was no evidence that such a concentration could be reached intracellularly under physiological conditions. Two major findings completely changed our understanding of the roles of calpain-1 and calpain-2 in LTP. First, we discovered that brain-derived neurotrophic factor (BDNF), a neurotrophic factor critically involved in synaptic plasticity and in LTP [56], could activate calpain-2 though ERK-mediated phosphorylation at serine 50 [57]. Moreover, we later found that BDNF-induced stimulation of local protein synthesis, another critical process in LTP formation [58], was mediated by calpain-2 activation of mTOR [59]. Detailed analysis of this mechanism indicated that calpain-2, but not calpain-1, cleaved phosphatase and tensin homolog (PTEN), a negative mTOR regulator, thus suggesting that calpain-2 activation following LTP induction could lead to increased local protein synthesis and participate in LTP consolidation [59]. The other major finding that changed our understanding of the role of calpain in LTP was provided by results from experiments performed with small modifications of the original protocol that was used to demonstrate the role of calpain in theta burst stimulation (TBS)-induced LTP in acute hippocampal slices [60,61]. In these experiments, a non-selective calpain inhibitor, calpain inhibitor III, was applied before delivering the theta burst. However, when calpain inhibitor III was applied immediately after TBS, the magnitude of LTP was dramatically enhanced; treatment with calpain inhibitor III 1 h after TBS had no effect [5]. These surprising results led us to propose a completely new model for LTP induction and consolidation, at least in field CA1 of the hippocampus. In our model, calpain-1 activation following synaptic NMDA receptor stimulation results in PHLPP1β degradation and ERK activation. However, calpain-2 activation following TBS, possibly resulting from BDNF-mediated ERK activation extrasynaptically, leads to PTEN truncation and mTOR activation followed by the stimulation of local protein synthesis and, in particular, of PHLPP1β synthesis, which would limit the duration of ERK activation [5]. To validate the model, we tested the effects of a selective calpain-2 inhibitor, the dipeptide ketoamide, Z-Leu-Abu-CONH-CH2-C6H3[3,5-(OMe)2] (C2I), which exhibits a more than 20-fold selectivity for calpain-2 over calpain-1, on LTP induction when applied before or after TBS [5]. The results were unambiguous, as C2I application either before or 10 min after TBS produced the same enhancement in LTP magnitude as when calpain inhibitor III was applied after TBS.

The differential effects of calpain-1 and calpain-2 on SCOP/PHLPP1β could be due to a different subcellular localization of the proteases. In particular, a fraction of PHLPP1β has been shown to be present in membrane rafts, and this fraction could be degraded following NMDA receptor channel opening and the resulting calpain-1 activation. On the other hand, calpain-2 might be present in a different subcellular compartment, and in particular, at the base of dendritic spines where local dendritic protein synthesis is taking place. Thus, the biphasic effect of TBS on ERK activation, i.e., initial activation followed by delayed prevention of activation, could reflect the sequential activation of calpain-1 and calpain-2 following TBS, as well as their opposite effects on PHLPP1β levels and ERK activation. An alternative, although not exclusive, hypothesis is suggested by the role of BDNF on calpain-2 activation through ERK-mediated phosphorylation. The release of BDNF during TBS, as evidenced by the activation of its TrkB receptor at synapses, is known to facilitate the cytoskeletal changes required for LTP consolidation [62,63]. In addition, the effect of BDNF on calpain-2 activation indicates that, by stimulating local protein synthesis and in particular the synthesis of SCOP/PHLPP1β, it triggers ERK inhibition, thereby limiting the magnitude of the LTP. Further studies using mice with a deletion of calpain-2 in hippocampal pyramidal neurons will verify the critical role of calpain-2 in limiting LTP magnitude following TBS. Further studies will also be directed at evaluating the potential role of calpain-2 in other forms of synaptic plasticity at various synapses. Likewise, further studies will determine whether calpain-2 activation also participates in different types of learning and memory.

## 5. Calpain-2 Role in Neuronal Death

While an extensive literature links calpain activation with neurodegeneration, very few studies have explored the specific contributions of calpain-1 and calpain-2 in neurodegeneration. As discussed above, we demonstrated that calpain-1 activation is neuroprotective both during the postnatal period and in the adult. This conclusion would therefore lead to the hypothesis that it is calpain-2 that is responsible for the reported role of calpain in neurodegeneration, unless some other calpain isoform would perform this function. Using primary neuronal cultures, we first showed that calpain-2, but not calpain-1 activation was responsible for NMDA-induced excitotoxicity through the activation of Striatal-Enriched Protein Tyrosine Phosphatase (STEP) [48]. A similar study indicated that the downregulation of calpain-2 but not calpain-1 increased neuronal survival following NMDA treatment of cultured hippocampal neurons [64]. It has been proposed that the activation of synaptic and extrasynaptic NMDA receptors have opposite effects on neuronal survival and degeneration [65]. Our hypothesis is that calpain-1 activation is directly downstream of synaptic NMDA receptors, while calpain-2 activation results from stimulation of extrasynaptic NMDA receptors. These receptors are enriched in NR2B subunits [66], which bind RasGRF1, thereby providing a link between NMDAR activation and ERK activation [67]. This pathway could therefore be responsible for the prolonged activation of calpain-2 following the stimulation of extrasynaptic NMDA receptors. In addition, calpain cleaves striatal-enriched tyrosine phosphatase (STEP), resulting in the activation of p38 and downstream cell death signaling pathways [68,69].

Other pathways could link calpain-2 activation to cell death. A recent study demonstrated that calpain-2 inhibition or knock-down enhanced autophagy and reduced cell death after ischemia/reperfusion in liver [70]. Similarly, calpain inhibitors promoted mTOR-independent autophagy and rescued Huntington’s disease phenotypes in zebrafish [71]. In Alzheimer’s disease (AD), calpain-2 was found to be hyper-activated in synaptosomes in presymptomatic AD, and the activation was correlated with a decline in cognitive function and an increase in levels of β-amyloid deposits [72]. Calpain activation has also been shown to switch cellular programs from autophagy to apoptosis in various preparations [73,74,75]. Likewise, calpain activation has been repeatedly shown to stimulate apoptosis pathways through multiple mechanisms [76]. More recently, we discovered that calpain-2 could cleave and inactivate the protein tyrosine phosphatase, PTPN13, aka Fas-associated protein-1 (FAP1) [77]. This phosphatase is an inhibitor of apoptosis, and therefore, by inactivating it, calpain-2 activation would stimulate apoptosis. Calpains also cleave several members of the Bcl-2 family of proteins, including Bax, Bid, and Bcl-xL, leading to cytochrome c release [78,79,80] and caspase-3 activation [81]. Calpain also converts pro-caspase-7 to caspase-7 [82]. Therefore, many studies indicate that calpain-2 activation prevents autophagy and stimulates apoptosis. Consequently, it is highly likely that calpain-2 activation represents a critical step towards cell death.

We previously identified another mechanism linking calpain activation to neuronal death through the truncation of mGluR1α, following NMDA receptor stimulation-induced calpain activation [83]. Under control conditions, mGluR1α receptors are coupled to PI3K-Akt signaling and their activation is neuroprotective. Brief mGluR1α activation leads to calcium release from internal stores, but the extent of calcium release does not produce significant toxic effects. Following NMDA receptor stimulation or ischemia onset, calpain activation leads to mGluR1α truncation, disrupting the neuroprotective effect of the mGluR1α-PI3K-Akt signaling cascade. Importantly, truncated mGluR1α receptors maintain their stimulation of calcium release from intracellular stores, which further contributes to calcium overload through NMDA receptors and enhances neurotoxicity [83]. 

Calpain activation has long been shown to be involved in the pathology of traumatic brain injury (TBI) [84,85,86]. Calpain activation results in the truncation of brain spectrin and several spectrin breakdown products (SBDPs) have been used as biomarkers for TBI in both cerebro-spinal fluid (CSF) and blood [87,88,89,90,91,92,93,94]. Surprisingly, none of these studies has addressed the respective roles of calpain-1 and calpain-2 either in brain pathology or in the generation of the blood biomarkers. This is likely due to the lack of isoform-selective calpain inhibitors, the lack of animals with selective deletion of calpain-1 or calpain-2, and the lack of markers for calpain-1 and calpain-2 activation. These limitations could account for several conflicting results. In particular, in some studies, the calpain inhibitors AK295 and ALLM were reported to protect the cytoskeletal structure of injured neurons and to attenuate motor and cognitive deficits after TBI [95,96]. In other studies, two other calpain inhibitors, SNJ-1945 and MDL-28170, which were shown to cross the blood–brain barrier, did not exhibit significant efficacy in a model of controlled cortical impact [97,98]. 

As mentioned above, using global calpain-1 KO mice, we established that calpain-1 is neuroprotective in a mouse model of TBI, the controlled cortical impact (CCI). To address the question of the roles of calpain-2 in both the pathology and the production of the blood biomarker, we used mice with selective deletion of calpain-2 in excitatory neurons of the forebrain, or selective calpain-2 inhibition provided by injection of C2I. Selective deletion of calpain-2 in excitatory neurons of the forebrain was obtained by crossing loxP-calpain-2 mice with CmKII-Cre mice, producing calpain-2 conditional KO (C2CKO) mice; immunohistochemistry confirmed that calpain-2 was significantly deleted in the cerebral cortex and in hippocampus, including field CA1 (Figure 1). Changes in SBDP levels in the cortex after TBI represent both calpain-1 and calpain-2 activation, as spectrin can be cleaved by both calpain-1 and calpain-2 [99]. However, by comparing the changes in SBDP at various times after TBI in wild-type (WT) mice and calpain-1 KO mice, we could estimate the respective contributions of calpain-1 and calpain-2 activation in SBDP generation [52]. Moreover, since PTEN is selectively cleaved by calpain-2 but not calpain-1 [59], analyzing changes in PTEN at various times after TBI also reflects the temporal activation of calpain-2 after TBI. Our results indicated that calpain-1 is rapidly and transiently activated in the cortex surrounding the impact site after TBI, with a maximal activation 6 h after TBI, but was no longer activated 24 h after TBI. In contrast, calpain-2 activation was delayed, starting between 4 and 8 h after TBI and was still present 3 days after TBI [52]. Western blots from cortical homogenates confirmed that 24 h after TBI, the increase in the levels of the 145–150 Kd SBDP fragments was similar in WT and calpain-1 KO mice but almost completely absent in C2CKO mice (Figure 2). Additionally shown in Figure 2 are western blots showing that the truncation of STEP61 resulting in the formation of the neurodegenerative fragment STE33 is also completely mediated by calpain-2. Similarly, the truncation of PTPN13 and the formation of breakdown products, labeled P13BPs, is entirely mediated by calpain-2 activation. Interestingly, in this model, we did not detect the truncation of TDP43, a protein known to be involved in Amyotrophic Lateral Sclerosis (ALS), fronto-temporal dementia (FTD), and repeated concussions [100,101]. Results from immunohistochemistry confirmed that the large increase in SBDP in the cortex surrounding the lesion site 24 h after TBI was entirely due to calpain-2 activation, as it was absent in C2CKO mice (Figure 3). Moreover, the large decrease in PTEN levels was also absent in C2CKO mice (Figure 3). Using Terminal deoxynucleotidyl transferase dUTP nick end labeling (TUNEL) and Fluoro-Jade staining to identify dying cells, we found that the extent of calpain-2 activation was positively correlated with the extent of cell death [52]. Systemic administration of C2I, 1 or 4 h after TBI, significantly reduced calpain-2 activation and the number of degenerating cells in the cortex surrounding the impact site, further demonstrating the neurodegenerative role of calpain-2. In good agreement with this result, the lesion volume 7 days after TBI was significantly smaller in C2CKO mice than in control mice (Figure 4). Finally, we also determined the levels of calpain-cleaved αII-spectrin N-terminal fragment (SNTF), a blood biomarker shown to be present in human subjects after concussion [92,93,94], in the blood of mice subjected to TBI with or without treatment with a selective calpain-2 inhibitor (Figure 5). First of all, we found that SNTF blood levels were very well correlated with the levels of calpain-2 activation in the brain; in addition, treatment of the mice with C2I 1 h after TBI prevented the increase in blood SNTF elicited by TBI. In a recent study, we also observed that calpain-2 deletion or semi-chronic treatment with a selective calpain-2 inhibitor prevented all the pathological manifestations resulting from repeated concussions in mice [101].

The same pattern of a brief activation of calpain-1 and delayed but chronic activation of calpain-2 was also found in retinal ganglion cells after retinal ischemia/reperfusion injury, a mouse model of primary angle-closure glaucoma (PACG) [53]. All these results are consistent with the idea that calpain-2 activation is delayed and prolonged following acute insults or repeated concussions. These results suggest that calpain-2 activation is responsible for the neurodegeneration as well as the brain inflammation resulting from brain insults. Calpain-2 activation may also be responsible for the cognitive impairment associated with the neuropathology observed weeks and months after the insults.

## 6. Conclusions

The long journey that started 40 years ago when we were trying to understand the role of calpain in synaptic plasticity and learning and memory has led us to explore a vast array of pathways and mechanisms. It has also uncovered the remarkable finding that the two major isoforms of calpain in the brain, calpain-1 and calpain-2, play opposite functions in many of these mechanisms. In particular, our studies revealed that calpain-1 activation is critical to trigger various types of synaptic plasticity in both hippocampus and cerebellum, which are engaged in various types of learning. Furthermore, calpain-1 activation is neuroprotective both during the postnatal development and in adulthood. Interestingly, several human families with null mutations in CANP1 have been identified and shown to exhibit profound neurological disorders, and in some cases cognitive impairment [49]. In contrast, calpain-2 activation limits the extent of synaptic plasticity and of learning during a brief consolidation period lasting about 1 h. The signaling pathways underlying these opposite functions of calpain-1 and calpain-2 have started to be identified and we have postulated that they are due to the association of calpain-1 and calpain-2 with different PDZ binding domains, resulting in their linkages to different signaling cascades [14]. These opposite functions are schematically represented in Figure 6, in which we stress the point that calpain-1 is preferentially linked to synaptic NMDA receptors while calpain-2 is proposed to be downstream of extra-synaptic NMDA receptors. Even more importantly, prolonged calpain-2 activation following a variety of acute brain insults is responsible for the neuronal damage resulting from these insults. It is also responsible for the brain inflammation associated with such insults as well as the long-term cognitive impairments produced by these insults. The identification of selective calpain-2 inhibitors that can prevent the neuropathological manifestations of these acute insults provide the hope that it will be possible to translate the animal results into clinical applications.

## Figures and Tables

**Figure 1 cells-09-02698-f001:**
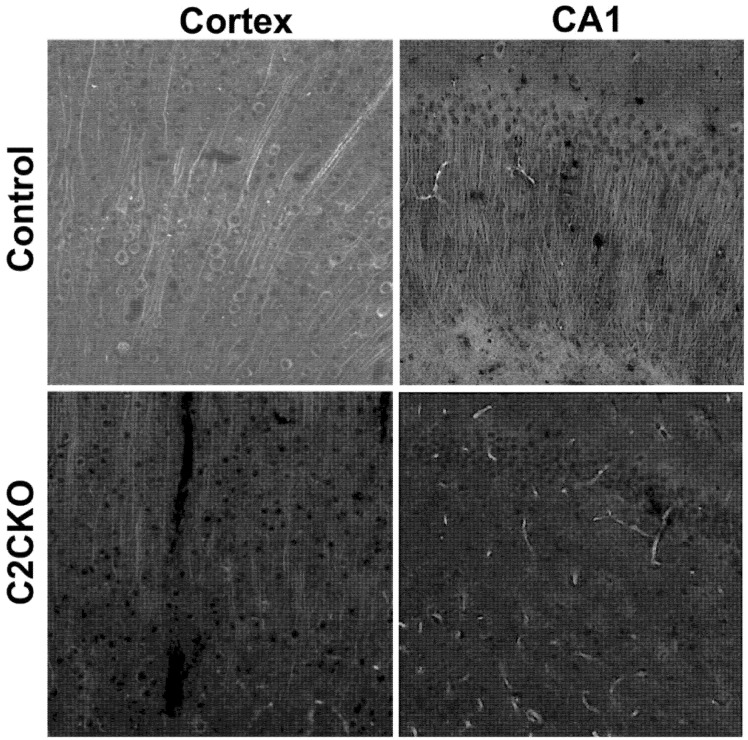
Calpain-2 deletion in cortex and hippocampal field CA1 in calpain-2 conditional knock-out (C2CKO) mice. Immunohistochemistry (IHC) with calpain-2 antibody (1:300; LS-C337641, LSBio) in cortex and CA1 of control (calpain-2 loxP) and C2CKO mice. Note the very large decrease in calpain-2 immunoreactivity in cortex and field CA1 of hippocampus. Scale bar: 50 µm. Details for the methods can be found in the Appendix A.

**Figure 2 cells-09-02698-f002:**
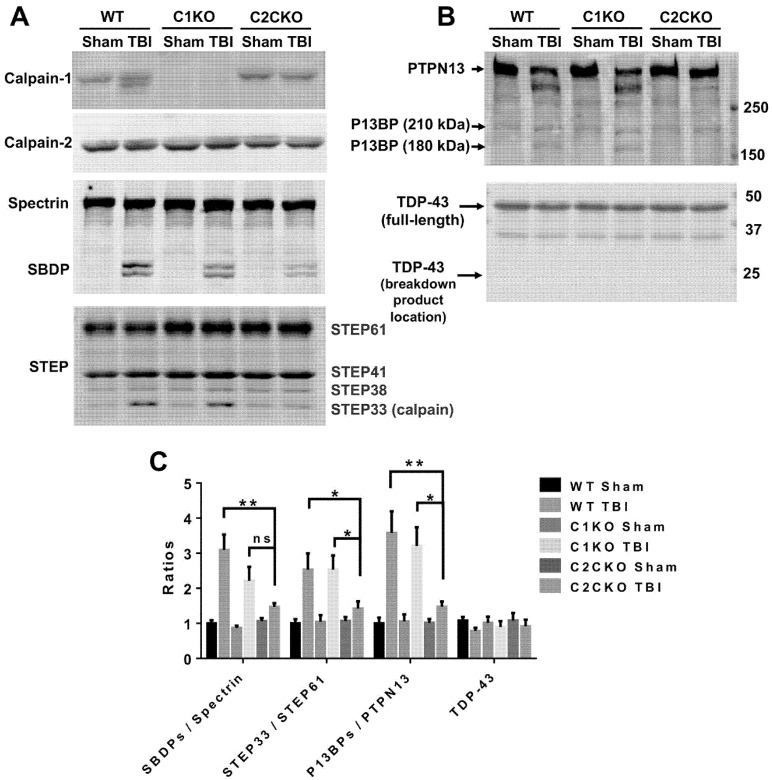
Changes in spectrin, striatal-enriched tyrosine phosphatase (STEP), PTPN13 breakdown product (P13BP), and TAR DNA-binding protein of about 43 kDa (TDP-43) in brain of WT, calpain-1 knock out (C1KO), and C2CKO mice 24 h after traumatic brain injury (TBI). (**A**,**B**) Representative western blots (WBs) of ipsilateral cortical tissue (P2 fraction) from WT, C1KO, and C2CKO mice collected 24 h after TBI or sham surgery. Note that calpain-1 is absent in C1KO mice and calpain-2 levels were reduced in C2CKO mice. Several calpain substrates including spectrin (1:500; MAB1622, EMD Millipore, Burlington, MA, USA), STEP (1:1000, NB300-202, Novus Biologicals, Colorado, CO, USA), PTPN13 (1:1000, PA5-72906, Thermo Fisher Scientific, Waltham, MA, USA), and TDP-43 (1:1000, 10782-2-AP, Proteintech, Chicago, IL, USA) were probed, and their fragments generated by calpain cleavage including SBDP, STEP33, and P13BPs were detected after TBI. (**C**) Quantification of WB. Ratios of SBDPs to spectrin, STEP33 to STEP61, P13BPs to PTPN13 were significantly increased in WT and C1KO but not in C2CKO mice after TBI. Levels of TDP-43 were not changed after TBI in all genotypes. * *p* < 0.05, ** *p* < 0.01. ns, no significant difference. N = 4. One way ANOVA followed by Bonferroni’s test. Full immunoblots for the blots shown here are presented in the Appendix A.

**Figure 3 cells-09-02698-f003:**
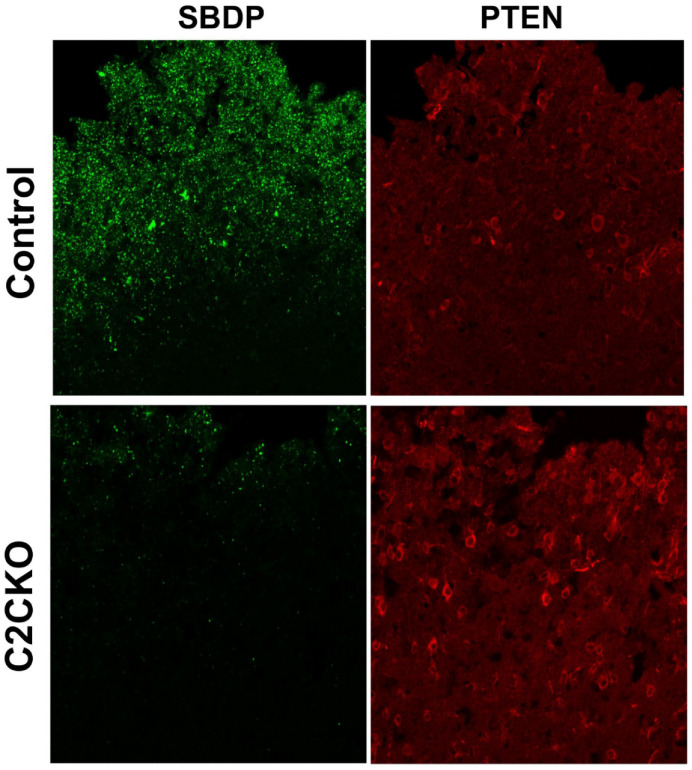
Changes in spectrin breakdown product (SBDP) and Phosphatase and Tensin Homolog (PTEN) around the lesion site in control and C2CKO mice after TBI. IHC with antibodies targeting SBDP (1:500, a gift from Dr. Saido, Riken, Japan) and full-length PTEN (1:600, 9556, Cell Signaling) around the lesion site was performed in control and C2CKO mice 24 h after TBI. Note that the levels of SBDP were much lower while PTEN levels were much higher in C2CKO mice as compared to control mice. Scale bar: 50 µm.

**Figure 4 cells-09-02698-f004:**
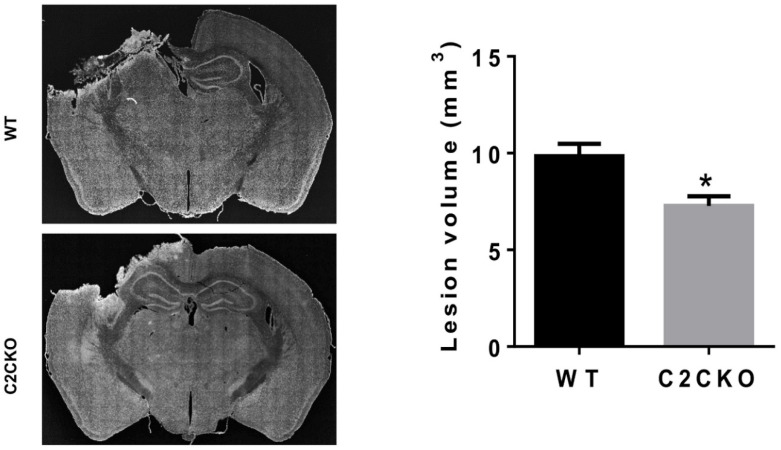
Reduced lesion volume after TBI in C2CKO mice. (**Left**): Representative images of Nissl-stained brain sections (Bregma 1.58 mm) in WT and C2CKO mice collected 3 days after TBI. Scale bar: 1 mm. (**Right**): Lesion volume in WT and C2CKO mice measured 3 days after TBI. Lesion areas were measured in eight sections (Bregma 1.54, 0.50, −0.58, −1.58, −1.94, −2.30, −2.70, and −3.40 mm) from each brain. Total lesion volume in each brain was calculated based on the lesion area in each section and the distance between sections. N = 5 for WT, N = 4 for C2CKO. * *p* < 0.05, *t*-test.

**Figure 5 cells-09-02698-f005:**
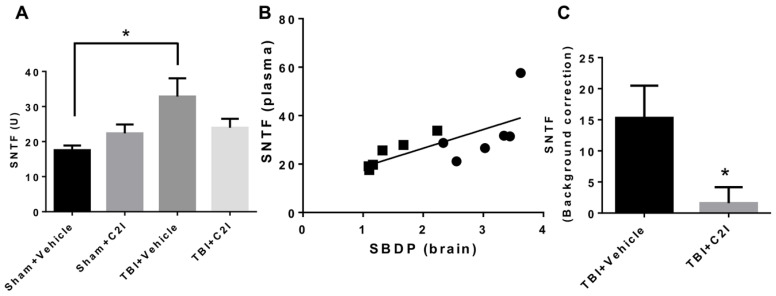
Levels of Spectrin N-Terminal Fragment (SNTF) in mouse blood 24 h after TBI with or without treatment with calpain-2 inhibitor (C2I). (**A**) Levels of SNTF in plasma of CD-1 mice collected 24 h after TBI. C2I (0.3 mg/kg) or vehicle was injected intravenously 1 h after TBI. * *p* < 0.05. N = 4 for Sham + Vehicle and Sham + C2I. N = 6 for TBI + Vehicle and TBI + C2I. One way ANOVA followed by Bonferroni’s test. (**B**) Correlation between SNTF levels in plasma and SBDP levels in brain of CD-1 mice 24 h after TBI. Black circles: TBI + Veh; black rectangles: TBI + C2I. R = 0.70. (**C**) Increase in SNTF levels after TBI in plasma of CD-1 mice collected 24 h after TBI. SNTF levels after sham surgery (background SNTF level) were subtracted from SNTF levels after TBI. * *p* < 0.05. N = 6, *t*-test.

**Figure 6 cells-09-02698-f006:**
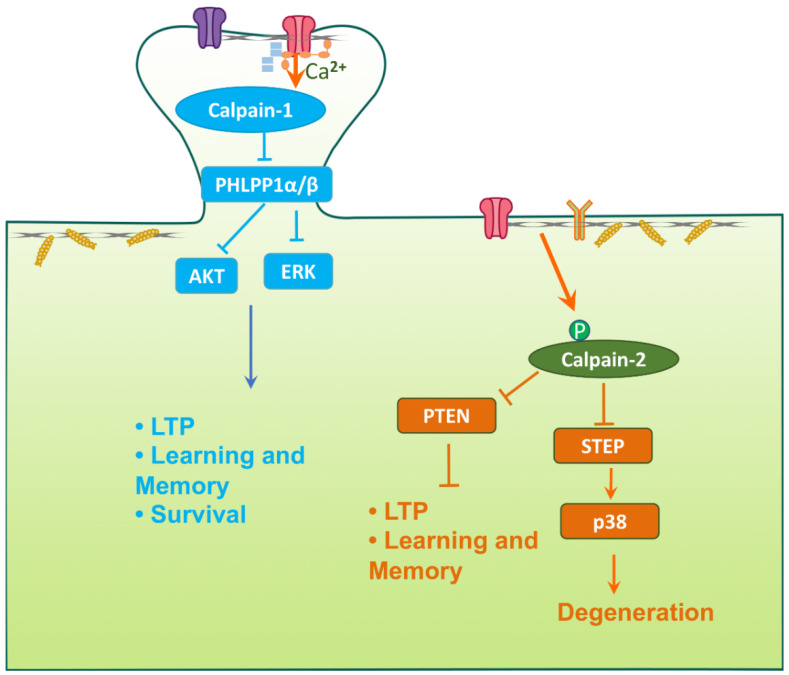
Different subcellular locations and functions of calpain-1 and calpain-2 in excitatory neurons. Calpain-1 is preferentially linked to synaptic N-Methyl-D-Aspartate (NMDA) receptors while calpain-2 is downstream of extra-synaptic NMDA receptors in neurons. Synaptic NMDAR activation activates calpain-1, which cleaves and inhibits PHLPP1α/β. PHLPP1α/β inhibits AKT and ERK. Thus, calpain-1-mediated cleavage of PHLPP1 activates AKT and ERK, which trigger long-term potentiation (LTP) and promote neuronal survival. On the other hand, extra-synaptic NMDAR activation induces phosphorylation and prolonged activation of calpain-2. Calpain-2 cleaves its substrates such as PTEN and STEP, leading to reduced LTP magnitude and neurodegeneration.

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
