# Peer review of "Calpain-1 and Calpain-2 in the Brain: New Evidence for a Critical Role of Calpain-2 in Neuronal Death"

_cells, 2020, doi:10.3390/cells9122698_

Round 1

Reviewer 1 Report

General Comments

There are five figures of what appears to be new data supporting a role of Calpain 2 in neurodegeneration. This is not appropriate for a review. This data should be removed, or the paper should be submitted as a research article.

The abstract talks about trying to explain differences between calpain 1 and calpain 2 functions (i.e. distinct PDZ binding sites), but this is not a part of the review and the specific sites and binding partners are not discussed. This discussion would be a nice addition to the review.

Specific suggestions

Line 32, “While generally ignored by the scientific community”; This kind of statement does not belong in a review. This provides no additional information to the review.  

Line 117; Stargazin is not used presently for this family; the more accepted name is transmembrane AMPA regulatory protein (TARP). The isoform of TARP discussed should be given.

Line 119; There is strong evidence that Granule Cell-Purkinje Cell LTD is not required for classical conditioning (Schonewille et al, Neuron. 2011 Apr 14;70(1):43-50.). This needs to be discussed. 

Line 152.  It is not correct to call calpains in invertebrates, Calpain-1. Calpain 1 and 2 diverged quite recently; indeed all the classical calpains in vertebrates probably diverged from one or two calpains present in early deuterostomes. It is more correct to call invertebrate calpains, just classical calpains, not Calpain-1. The calpain in Drosophila and C. Elegans is clearly not more similar to Calpain 1 than to Calpain 2.

Line 172-186. If BDNF-induced protein synthesis is dependent on Calpain-2 activation of TOR, (line 165-169), then how can LTP be enhanced in the absence of Calpain-2 (rest of paragraph).  The role of protein synthesis in generating inhibitors of LTP is not reconciled with the requirement of protein synthesis for LTP. This section needs a better discussion of this somewhat contradictory logic.

I would discourage references to papers in preparation in reviews; at least an abstract should be referenced (Lines 203-204).

Line 344-346. These statements are made as facts, but I think they are suggestions based on the data provided. IF they are facts, then references would be required to back up these statements. For example, there are no congnitive tests mentioned in the data presented, and if these experiments have been done, a reference is required for the statement.

Lines 354 to 356. The descriptions of the humans with CAPN1 mutations talked about motor diseases, ataxia and paraplegia. There is no real description of cognitive impairments, and if so, the specific papers that describe it and the percentage of patients with cognitive impairments should be discussed.

Figure 6. The figure has ERK on the LTP side and calpain 1, but the mechanism for activation of Calpain 2 is supposedly ERK-mediated. This is confusing.

The abstract talks about trying to explain differences between calpain 1 and calpain 2 functions (i.e. distinct PDZ binding sites), but this is not a part of the review and the specific sites and binding partners are not discussed. This discussion would be a nice addition to the review.

Reviewer 2 Report

“Calpain-1 and calpain-2 in the brain: Major roles in 2 synaptic plasticity and neuronal life and death” is a review of calapain in synaptic plasticity and in neurodegeneration. The review is well-written and focuses on the opposite roles for Calpain-1 and Calpain-2.  The authors provide an excellent review of the literature that details how the major discoveries have led to their current understanding for these enzymes.  This review provides enough background information in order for neuroscientists in different areas of expertise to appreciate, while maintaining the scientific integrity of the research.  It’s a difficult task to balance but these authors have done it well!

There are a few minor points:

  1. Consider revising sentence L88-91.  Awkward.
  2. SNTF needs to be spelled out (L296).
  3. Fig 2A:  Calpain-2 levels do not appear reduced in the western blot for C2CKO
  4. Consider adding the F values in the fig caps for the ANOVAs.

Reviewer 3 Report

Overall, a comprehensive and well written review on the role of calpain 1 and 2 in synaptic plasticity and neuronal death by the Baudry Lab. I have no changes to propose, the manuscript is sound and a very good read, figures are correct and clear with adequate legends. I recommend acceptance as is.

Reviewer 4 Report

This is a well written manuscript which does have a format of a review but it is a mix of a review and a research article. The title is not appropriate as the manuscript is mostly dealing with calpain 2 in traumatic brain injury while a broad function of both calpains to different diseases is not discussed including other research studies being not cited. There are studies investigating calpain 1 and calpain 2 in Alzheimer’s disease e.g. Ahmad et al., scientific reports 2016 (PMID: 30177812).

The novel data presented here have to be reviewed in detail but this is difficult as the classical material and methods section and experimental/ statistical detail are missing. Below are my major comments related to the new data

  1. Page 6, Line 269. The authors state a significant deletion of calpain 2 in KO mice and refer to figure 1. However, no quantifications are provided in this figure and the images in black and white do not allow a good appreciation of the protein levels. The authors should include a quantification measure and/or high-quality images (preferably using different colours for neuronal marker and calpain 2 proteins).
  2. Page6, Line 285. The inclusion of TDP43 and the reference as it is not changed here is out of context as the reference article did report its truncation in ALS while the data shown here are related to TBI. The authors do not show conditions were TDP43 is cleaved to be able to estimate the contribution of the different calpains in their respective KO animals.
  3. Page 6, Line 287 and Figure 3. The authors state that spectrin breakdown products (SBDP) are increased after 24h and that this increase is largely due to calpain 2 since this increase was not observed in Calpain KO mice and they refer to immunohistochemistry data from figure 3. First, I suppose that there is no specific antibodies that detect only the spectrin breakdown products and that the signal observed in figure 3 corresponds to both spectrin as well as spectrin breakdown products. Therefore, it’s not correct to refer to them as spectrin breakdown products. Second, the data shown in figure 2 does not support an increase in spectrin signal after TBI at the same time 24h. I suppose that the signal observed in Figure 3 is largely due to cell death at the legion site and an abundance of extracellular spectrin. Cell death might be attenuated in calpain2 KO mice which would explain the reduced signal of spectrin observed in Figure 3.

Round 2

Reviewer 1 Report

No comments

Author Response

This reviewer has no comments.

Reviewer 4 Report

The authors have addressed all my concerns. I have no further comments

Author Response

This reviewer has no comments.